# Navigation aid for blind persons by visual-to-auditory sensory substitution: A pilot study

**Alexander Neugebauer**[1]*, **Katharina Rifai**[1,2], **Mathias Getzlaff**[3], **Siegfried Wahl**[1,2]

**1** ZEISS Vision Science Lab, Eberhard-Karls-University Tuebingen, Tübingen, Germany, **2** Carl Zeiss Vision International GmbH, Aalen, Germany, **3** Institute for Applied Physics, Heinrich-Heine University Duesseldorf, Duesseldorf, Germany

* a.neugebauer@uni-tuebingen.de

## Abstract

### Purpose

In this study, we investigate to what degree augmented reality technology can be used to create and evaluate a visual-to-auditory sensory substitution device to improve the performance of blind persons in navigation and recognition tasks.

### Methods

A sensory substitution algorithm that translates 3D visual information into audio feedback was designed. This algorithm was integrated in an augmented reality based mobile phone application. Using the mobile device as sensory substitution device, a study with blind participants (n = 7) was performed. The participants navigated through pseudo-randomized obstacle courses using either the sensory substitution device, a white cane or a combination of both. In a second task, virtual 3D objects and structures had to be identified by the participants using the same sensory substitution device.

### Results

The realized application for mobile devices enabled participants to complete the navigation and object recognition tasks in an experimental environment already within the first trials without previous training. This demonstrates the general feasibility and low entry barrier of the designed sensory substitution algorithm. In direct comparison to the white cane, within the study duration of ten hours the sensory substitution device did not offer a statistically significant improvement in navigation.

## Introduction

Many of our everyday tasks and activities rely on the sense of vision. In fact, visual information has been estimated to sum up to more than 99% of all the information perceived by the human senses [1]. Although only a rough estimation, this number gives an impression of the

**Data Availability Statement:** All relevant data are within the paper and its Supporting Information files.

**Funding:** Funding received from University of Tuebingen (ZUK 63) as part of the German

Excellence initiative from the Federal Ministry of Education and Research – Germany (BMBF). This work was done in an industry-on-campus-cooperation between the University of Tuebingen and Carl Zeiss Vision International GmbH. The authors received no specific funding for this work. The funder provided support in the form of financial compensation for participants but did not have any additional role in the study design, data collection and analysis, decision to publish, or preparation of the manuscript.

**Competing interests:** Work of authors is supported by the Institutional Strategy of the University of Tuebingen (Deutsche Forschungsgemeinschaft, ZUK 63). This work was done in an industry-on-campus-cooperation between the University of Tuebingen and Carl Zeiss Vision International GmbH. Two of the authors, Katharina Rifai and Siegfried Wahl, are scientists at the University of Tuebingen as well as employees of Carl Zeiss Vision International GmbH. Their affiliation with Carl Zeiss Vision International GmbH had no influence in the study. There are no competing interests related to employment, consultancy, patents, products in development, or marketed products. We fully adhere to PLOS ONE policies on sharing data and materials.

importance of vision and at the same time of the severe restrictions that come with the loss of that sense.

Worldwide, it is estimated that almost 40 million people are considered blind and another 245 million people are living with severe visual impairments [2]. Yet, despite this large number of people affected and the massive effect that blindness or visual impairment has on their everyday life, only few advances in the field of vision rehabilitation for navigation and navigation assistance tools for the blind have reached widespread use in public within the last decades, with the white cane still being the most common assistance tool for the blind [3–5].

One of the approaches made towards the rehabilitation of visual perception, meaning the reduction of the restrictions originating from a lack of vision, is sensory substitution. The term was first introduced in 1969 by the neuroscientist Paul Bach-y-Rita [6] and describes a non-invasive process to translate sensory stimuli like brightness or color into stimuli of a different sense, for example volume and pitch. It is most often used to compensate for the loss of one sense and works in all cases where the neuronal function of the brain is unaffected by the damage causing this disability [7, 8]. The sensory information is translated by a sensory substitution device (SSD), which typically consists of three parts: The input device that captures information of the type to be translated, a processing unit that translates the information following a specified algorithm, and an output device that transmits the translated information to the user. As an example, a visual-to-auditory SSD would require a visual input device such as a camera, a processing unit like a smartphone or notebook, and an auditory output device such as headphones.

What sensory-substitution-based devices excel at in regard to other visual aids is its direct translation of information without any interpretation or summary of information done by the device [9]. Every change in the input signal leads to a distinct change of the output. Thus, sensory substitution can offer almost the same flexibility and natural feeling that the original sense would offer. It is possible that the substituting sense leads to very similar activities in the brain as the original sense would, even in cases of congenital sensory disabilities—a process called 'brain plasticity' [7, 10–12] or, in the context of sensory substitution, 'compensatory plasticity' [13]. It was shown that the substituting signals evoke an increased activity in the occipital cortex after sufficient training with a device that translates visual information into tactile or auditory stimuli [14–16]. This indicates that the output signal, despite being an auditory or tactile stimulus, is perceived and processed in the same way as a visual stimulus.

However, the capacity to process information was found to vary greatly between the different senses, with the visual sense being able to process manifolds the amount of information of every other sense [1, 17, 18], as can be estimated by the number of fibers and their spike rates in optic, auditory or haptic nerves [19–23]. This is a problem always faced when designing a visual-to-auditory or visual-to-tactile sensory substitution algorithm, as it requires to compromise between translating the visual information as direct and unprocessed as possible while at the same time filtering only the most important information [24, 25].

Over the years, several different approaches have been made towards the substitution of the visual sense using either the auditory or tactile sense [26–28]. Two of the most popular devices in that field are the visual-to-auditory SSDs 'the voice' [24, 29–31] and 'EyeMusic' [32, 33]. These devices translate the brightness of a captured image into sound volume, with the image being scanned and translated from left to right in 2-second-intervals. EyeMusic additionally uses different instrumental sounds for each color, allowing to perceive not only brightness, but color as well. However, the interpretation of the output signals of these types of SSDs is difficult and requires long training phases [34–37], especially for navigational tasks, where the perspective constantly changes while moving.

Some newer devices, such as the SSDs 'EyeCane' [38], 'Sound of Vision' (SoV) [39] or 'Array of Lidars and Vibrotactile Units' (ALVU) [40] use depth as their primary visual input. Using this information, they directly translate the visual distance into either tactile or auditory feedback.

The EyeCane is a hand-held device that captures depth in a single direction and translates the distance into vibration strength. This results in the device functioning very similar to a standard white cane, however, it offers the advantage of increased detection range of up to 5m [41].

The SoV uses a combination of stereo camera and infrared sensor, both worn on the forehead, to capture a depth information array of the scene and translate each point of the array into the volume of an auditory output signal [42]. The auditory output of the array happens simultaneously, resulting in a high update rate, which allows for an easier understanding of the 3D shape of the environment. The SSD however requires a multitude of peripheral devices–multiple cameras and depth sensors and a laptop carried in a backpack–which makes it not only impractical and inconvenient in mobile scenarios such as navigation, but also increases the cost of the device.

The ALVU consists of an array of seven infrared depth sensors fixed to a wearable belt that capture the distance to obstacles and surfaces in an ±70˚ horizontal and ±45˚ vertical angle in front of the user, translating the information into vibration strength of a vibratory motor array. This allows the perception of obstacles and surfaces in a horizontal arc, similar to the information gathered by sweeping motion with a white cane, and additionally allows to perceive obstacles at waist or head level. This SSD was found to have a low learning barrier and allowed navigation performance similar to a white cane. However, due to the amount of individual sensors and vibratory motors, the cost sums up to around 1300\$ [40]. Given that 89% of visually impaired people live in low- or mid-income countries [43], a low-cost solution for SSDs is crucial.

Thanks to recent development in the field of real-time tracking of environmental structures–driven by the advancements in augmented reality software–new strategies to design, benchmark and implement SSDs in everyday life are emerging and allow novel applications. This paper describes a spatial-information-based visual-to-auditory translation algorithm that has minimal hardware requirements and is intuitive and fast to learn. To speed up the design and the test of the translation algorithm, it is simulated using an augmented-reality-based smartphone application. The performance of this approach is evaluated in both navigation and object recognition in a psychophysical experiment with blind participants.

## Translation algorithm and SSD

The design of the translation algorithm was led by the focus on easy and intuitive interpretation of the output signal. This design principle ensures a low entry barrier for potential users, while still giving it the precision and flexibility that is offered by direct information translation [9]. The algorithm translates visual spatial information, meaning distance, horizontal and vertical position, into the auditory stimuli pitch and volume and the proprioceptive sense of head movement (see Fig 1(A)), as it is designed to be used with a head-mounted camera.

The choices for the individual translations with regard to intuitiveness and accuracy will be further addressed in the discussion.

To decrease the information load on the auditory system and by that make it easier to focus on and interpret individual sounds of the auditory output, the proposed translation algorithm only translates a single, centered column of visual image points as opposed to the full field of view (Fig 1(B)). When implemented in a head mounted SSD, it allows to rotate the head

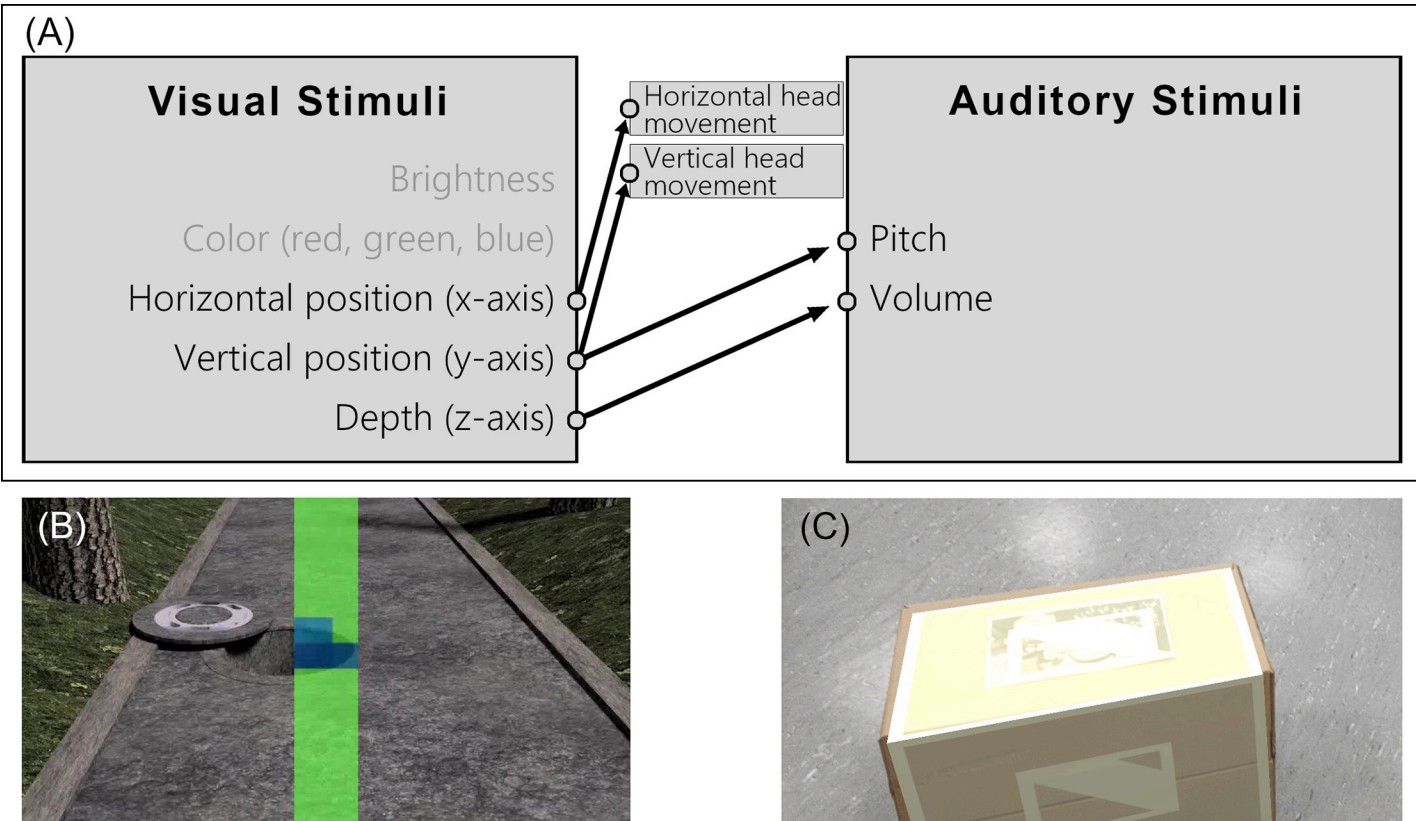

**Fig 1.** (A) Translation algorithm. Schematic visualization of the designed translation algorithm. (B) Translation algorithm visualization. Visualization of a translation algorithm translating the 3D information of the direction the user is facing. The color indicates the relative difference in height based on ground level. (C) Obstacle synchronization. A cardboard box used as an obstacle, overlaid with a virtual model that is synchronized to the real-world box using an image tracker.

horizontally in order to quickly scan the entire scene. To further reduce the auditory output, the algorithm suppresses sound output of visual image points whenever they do not show a change in height compared to ground level. This is visualized by the green area in Fig 1(B). Thus, the auditory output sound is generated only in situations in which the user faces an obstacle, reducing potentially distracting sounds to a minimum.

To test the newly designed translation algorithm, it was implemented in an augmented reality (AR) based SSD. Using the game engine Unity3D and the AR development kit ARCore [44], a software for Android devices was developed that is capable of full position- and rotation tracking. It allows the synchronization of a virtual environment with real-world obstacles (as shown in Fig 1(C)) and applies the designed translation algorithm to the virtual environment. The vertical angle of the visual input field was set to 45˚ to align with the vertical viewing angle of the smartphone camera. In the horizontal axis, the angle of the visual input equals 0˚ due to the single-column design of the translation algorithm (Fig 1(B)). The device tracks the distance to obstacles and surfaces in 9 different directions along the vertical axis, which results in a resolution of 5.6˚.

The information about the distance to the next surface measured by each individual ray is translated by the SSD into non-monotone sounds similar to burbling water. Each ray has its own respective pitch depending on its vertical direction, starting at around 143Hz for the lowest ray up to 880Hz for the highest.

As this SSD is aimed towards ease of access in terms of both training and hardware requirements, it was named Easy-Access SoundView (EASV).

## Experimental methods

### Ethics

Based on the above setup, an experiment was proposed and approved by the ethics committee of the faculty of medicine and the university hospital of the Eberhard-Karls-University Tuebingen in accordance with the 2013 Helsinki Declaration. All participants signed informed consent forms. The individual shown in Fig 2(A) of this manuscript has given written informed consent (as outlined in PLOS consent form) to publish these case details.

### Study population

Seven participants (5 female, 2 male) aged 19 to 75, average 33.1 ± 17.4, were recruited (Table 1). According to their self reports, all participants were considered blind or severely visually impaired following the ICD-10-system, having a visual acuity of 0.1 or below on both eyes [45].

All participants used the cane as the only direct navigation tool. Some of them occasionally use smartphone-apps with GPS-based navigation like Google Maps. None of the participants had any prior experience with an SSD.

The participants' hearing performance was measured in a simple up-down-method audiometry [46] to ensure that all participants are able to perceive the lowest volume in all frequencies of the auditory output of the EASV device. The audiometry used 0.5 second sound samples of five different frequencies (143Hz, 220Hz, 440Hz, 593Jz, 880Hz), covering the frequency range of the auditory output of the SSD. Each frequency was tested for 60 seconds, with randomized 4 to 10 second intervals between individual sound cues. After each correct

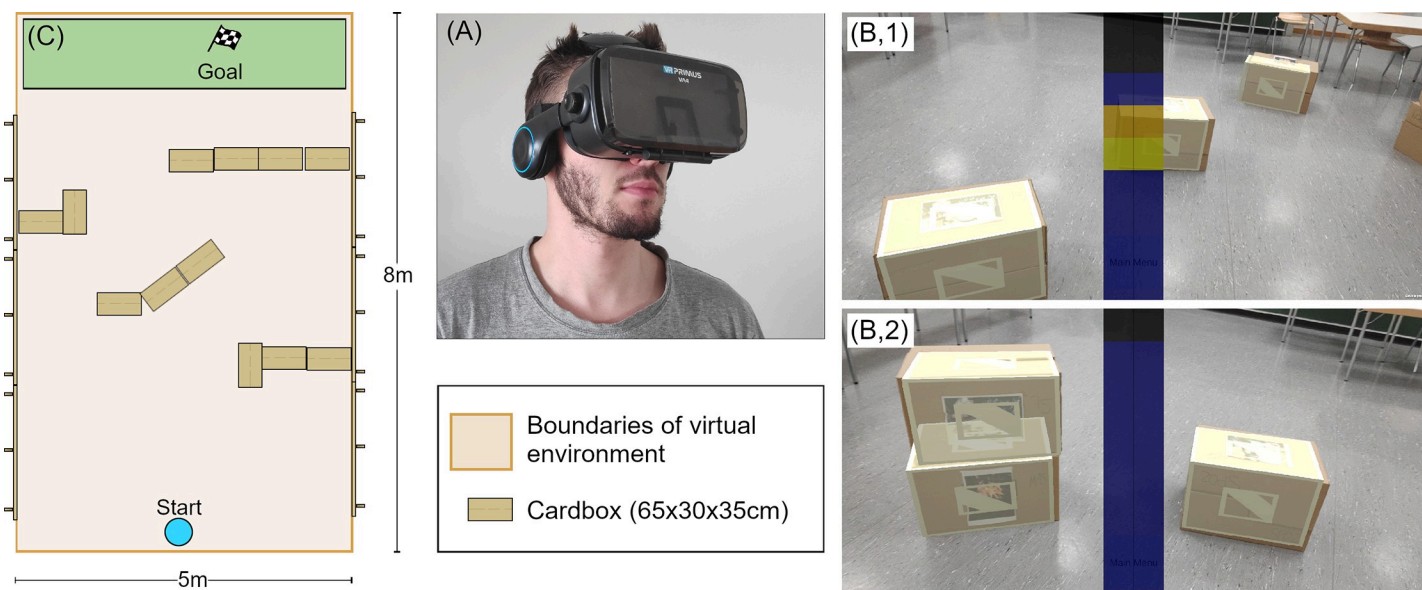

**Fig 2.** (A) Wearable device. A mobile VR-headset used to mount the smartphone to the head of the user. (B) Tutorial scenes. Cardboard boxes, overlaid with their virtual counterpart, in different tutorial scenes focused on distance (B1) and height (B2). The middle bar shows the visual representation of the auditory output, where dark blue/black indicates silence and yellow/red indicates a sound with the respective pitch. (C) Obstacle course. Layout of one of the 23 obstacle courses designed for the navigation task.

**Table 1. Study population.**

| Participant | Sex & Age | Duration of blindness | ICD 10 blindness category | Clinical picture | Hearing thresholds (143Hz, 220Hz, 440Hz, 593Hz, 880Hz) |
|---|---|---|---|---|---|
| 1 | m, 31 | ~6 years | H54.0X34 (VA = 0.03 on right eye, < 0.02 on left eye) | multiple scotoma on both eyes | 0.040, 0.0086, 0.0049, 0.0015, 0.0013 |
| 2 | f, 19 | ~3.5 years | H54.0X44 (VA < 0.02 on both eyes) | Myopia, Astigmatism, Tissue shrinkage of optical nerves | 0.0018, 0.0006, 0.0006, 0.0008, 0.0003 |
| 3 | m, 75 | since childhood | H54.0X55 (no light perception) | N/A | 0.015, 0.0032, 0.0013, 0.0049, 0.0015 |
| 4 | f, 25 | since birth | H54.0X44 (VA < 0.02 on both eyes) | N/A | 0.0028, 0.0022, 0.0015, 0.0026, 0.0022 |
| 5 | f, 29 | since birth | H54.0X44 (light perception) | Retinitis Pigmentosa, Myopia, Astigmatism, Nystagmus | 0.0058, 0.0038, 0.0038, 0.0026, 0.0015 |
| 6 | f, 27 | ~6 years | H54.1224 (VA = 0.08 on right eye, < 0.02 on left eye) | Cone-Rod Dystrophy, Glare sensitivity | 0.0086, 0.0056, 0.0015, 0.0022, 0.0015 |
| 7 | f, 26 | since birth | H54.0X44 (VA < 0.02 on both eyes) | N/A | 0.015, 0.0058, 0.0086, 0.0058, 0.0019 |

Detailed information about the study population. All data regarding duration of blindness, degree of blindness and clinical picture is based on participants' self report. The hearing threshold is given in form of a unit-less, software-internal value for sound volume. The threshold required to hear the lowest volume of the SSDs auditory output is 0.1.

response to the sound cue, the volume of the next cue was decreased by 33%. After each missed or false response, the volume was increased by 50%. After 60 seconds, the volume of the last recognized sound cue was saved and the test automatically proceeded with the next higher frequency. All volumes were measured in software-internal, unitless values, allowing the direct comparison to the auditory output of the SSD which uses the same software-internal values with a minimum volume threshold of 0.1. The results of the audiometry are listed in Table 1, with all results below the threshold of 0.1 indicating that the participant is able to fully perceive the auditory output in the respective frequency.

The audiometry was done in the same room and under the same noise conditions as the main experiments. There was no test for each individual ear as the auditory output of the SSD is monaural.

## Experimental setup

The per-participant study duration was set to ten hours, separated into five sessions (S1 Fig). The study was carried out in the timeframe of two months in three different seminar rooms of the Philipps-University Marburg, Germany. All rooms were well-lit and had a floor area larger than 60m$^2$. 12 cardboard boxes (0.65mx0.3mx0.35m) marked with distinct image tracking targets were used in the tutorial and navigation tasks. The camera footage captured by the SSD, combined with the overlaid virtual environment, was streamed to a laptop for easier supervision.

## SSD setup

By using a mobile virtual reality headset (Fig 2(A)) to fixate the smartphone used as SSD on the head, participants could move around freely without any wires or other constraints. This is a major advantage to the limited tracking space provided by most virtual reality headset setups. This choice also allowed to use real-world obstacles as opposed to a purely virtual setup, which increased immersion and representativeness.

## Participant tasks

In the tutorial, which took place in the first two-hour-session, participants were able to familiarize themselves with the headset and the translation algorithm. Different scenes were created

using the cardboard boxes (as seen in Fig 2(B)). Each scene focuses on a different part of the translation algorithm (distance, height, width) and by presenting the scenes to the participants, the meaning of the changes of the auditory output of the SSD was explained by the experimenter. The participants could inspect the scenes and touch the cardboard boxes, thus connecting the auditory output signals to their own tactile input.

The navigation tasks were divided into three conditions–navigation with SSD, navigation with cane and navigation with both SSD and cane combined. All trials were done in randomized order independent of their condition to minimize the influence of potential learning effects when comparing the trials of all three conditions. Each of the trial conditions has the same base structure: 12 cardboard boxes are positioned in an area of 5x8 meters according to one of different pre-designed layout maps (S2 Fig). An example is shown in Fig 2(C). The participant was directed to the starting point located at one side of the area, facing in the direction of the other side. They were then instructed to navigate through the area, avoiding the obstacles, until they reach the area marked as goal at the other side of the room. Time was recorded between the moment the participant started walking and the moment they arrived in the goal area. For each navigation trial a total duration of 7 minutes was estimated, including 4 minutes to restructure the obstacle course in-between trials. Depending on the average required time of the individual participant, 23–30 navigation trials were done per participant over the duration of the second to fifth session. The hardware setup of the SSD was worn during all navigation trials to measure time and collisions. Further, this method ensured that the results are not influenced by the remaining eyesight of the participants. In trials in which only the cane was used for navigation, the auditory output of the SSD was muted.

In the object recognition trials, participants were presented different virtual objects in individual, otherwise empty scenes (examples of these objects are found in Fig 3(A) and 3(B)), which they were asked to identify out of a selection of four different options using the SSD. The users could control at which point in time the object is no longer shown. Only after that they were allowed to answer. The goal of the object recognition trials was to find out if and to what degree recognition and correct interpretation of more complex shapes than the boxes used in the tutorial and navigation task is possible with the developed translation algorithm.

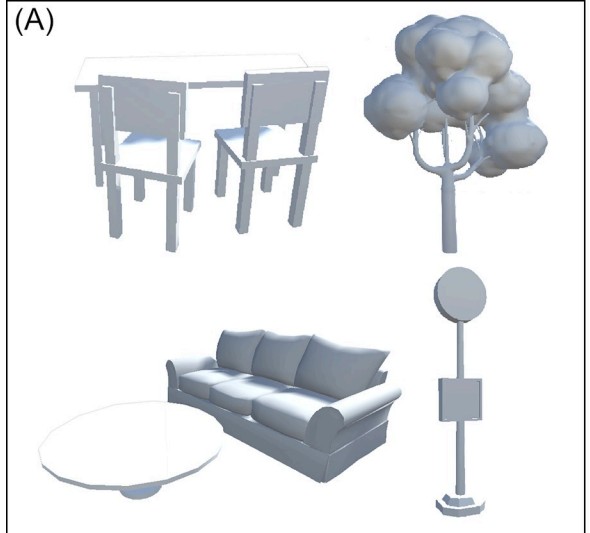
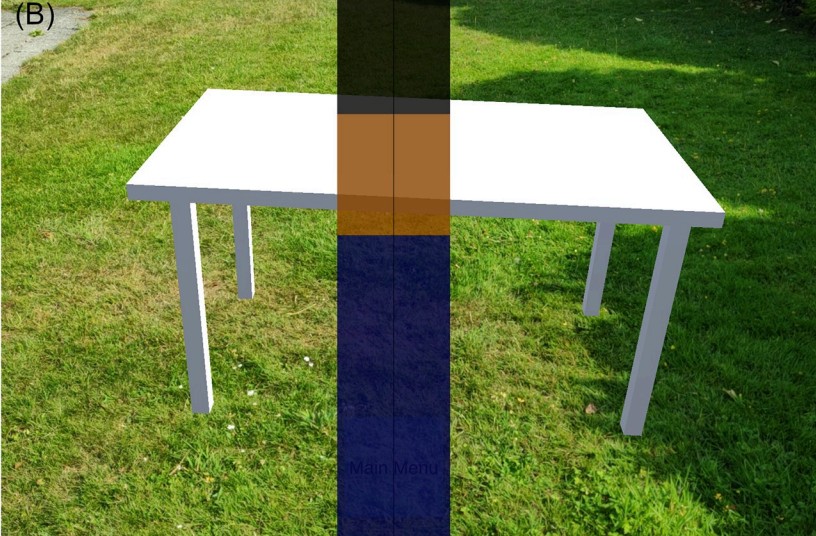

**Fig 3.** (A) Object recognition models. Examples of the 3D models used in the object recognition tasks. From left to right: Table with chairs; couch and couch table; tree; bus stop sign. (B) Position anchor. A virtual object displayed at a specific point in space using the augmented reality features of the software ARCore.

For the trial duration, an average of 3 minutes for each object was estimated and 14–16 trials were done per participant.

A concluding evaluation, comprised of five questions, assessed the learning rate, expected future learning rate and the potential of the SSD in different visual tasks and navigation environments, as subjectively perceived by the participants. It also served as means to collect final feedback in the form of freely answerable questions. The first part of the evaluation uses a structure based on a System Usability Scale [47], allowing participants to rate the different aspects on a scale from 1 (very low) to 5 (very high).

## Analysis and statistical methods

In the navigation task, a total of 190 trials (84 with SSD, 46 with cane, 60 with SSD+cane) in 23 different obstacle course layouts were done (S1 File). The time required to reach the goal area as well as the number of collisions were measured, with the collisions automatically being detected by the smartphone when its position overlapped with a virtual obstacle. The results for the required time are normalized using a logarithmic function. Linear mixed models (LMMs) were used to analyse the results of the conditions in which a learning effect is assumed (SSD and SSD+cane). LMMs allow to test for fixed effects between certain variables while still considering the random effects of other variables. The fixed effects are the correlations between trial condition (SSD, cane, SSD+cane) and trial number with both the required time for the trial and the number of collisions respectively, allowing to test for significant learning effects over the duration of the study in any of the conditions. Variables with random effects are all those that are assumed to have an influence on the results, however not in a predictable, ordered manner (participants, maps). For the trials with just the cane it was assumed that no correlation exists between required time and trial number. To test this, a Chi-squared test of independence was used.

In the object recognition tasks, the correctness of the answer given by the participants and the time during which the object was presented to them was measured in a total of 108 trials in 16 different scenes. A binomial test with the number of trials, the number of correct choices and the probability for each answer in an even distribution is used to analyse the results. The mean average and standard deviation of the required time in object recognition were calculated. The required time of 10 trials was not recorded due to either software issues or to participants not starting the timer before the trial.

All methods of analysis were done in the statistical computing environment R, using the graphical user interface RStudio. For the LMM analysis, the lme4-package was installed in R. All averages are shown as mean with the corresponding standard deviation.

The relatively low number of participants and thus low number of evaluations prevents a statistical analysis of correlations. Therefore, raw results of the evaluation are presented. One participant had to leave the last session early and was not able to fill the evaluation, leaving a sample size of six.

## Results

Fig 4 shows the normalized time needed for each navigation trial in relation to the trial number. Regression lines are drawn for each condition based on an LMM. To get an overview of the actual results of this task, the right y-axis of the graph additionally shows the actual time on a logarithmic scale.

With the SSD, the average time participants required to pass the obstacle course was 68.4 ±54.7s. This is significantly longer than the time required with either the cane (27.0±12.2s) or with both SSD+cane (37.3±29.7s) (p<0.001). While the independence between the navigation

**Fig 4. Required time for navigation trial.** Normalized time required per navigation trial in relation to the trial number, with a normalized linear regression line drawn for each condition.

time in trials with cane and the trial number is found to be valid (p = 0.0031), the null-hypothesis for the learning effects with SSD and SSD+cane could not be rejected (p = 0.36 with SSD, p = 0.12 with SSD+cane). Thus, no significant learning effect can be shown based on the required time for navigation. However, there is a tendency showing a performance increase for both the trials with SSD and SSD+cane.

With a mean average of 2.96±4.33 collisions, trials in which only the SSD is used are shown to have a worse navigation performance than trials in which a cane is used at an average 0.78 ±1.28 collisions (p<0.001). In trials with SSD+cane, the average number of collisions is almost identical to the trials with just the cane at 0.77±1.69. In the LMM, no significant performance increase could be found over the duration of the study (p = 0.48 for SSD trials, p = 0.094 for SSD+cane trials), but the tendencies of the regression again show a decrease of collisions only in trials in which the SSD is used (see Fig 5).

Table 2 shows the percentage of object recognition trials in which the object is correctly identified. The results of the binomial test for the data acquired in the object recognition task —a total of 108 trials and 58 correct object identifications in a setup where sets of four answers are given to the participants—confirm that the SSD does indeed lead to an identification rate significantly above the 25% threshold of even distribution (p<0.001). On average, participants spent 98±53.6s observing the object before being able to correctly identify it.

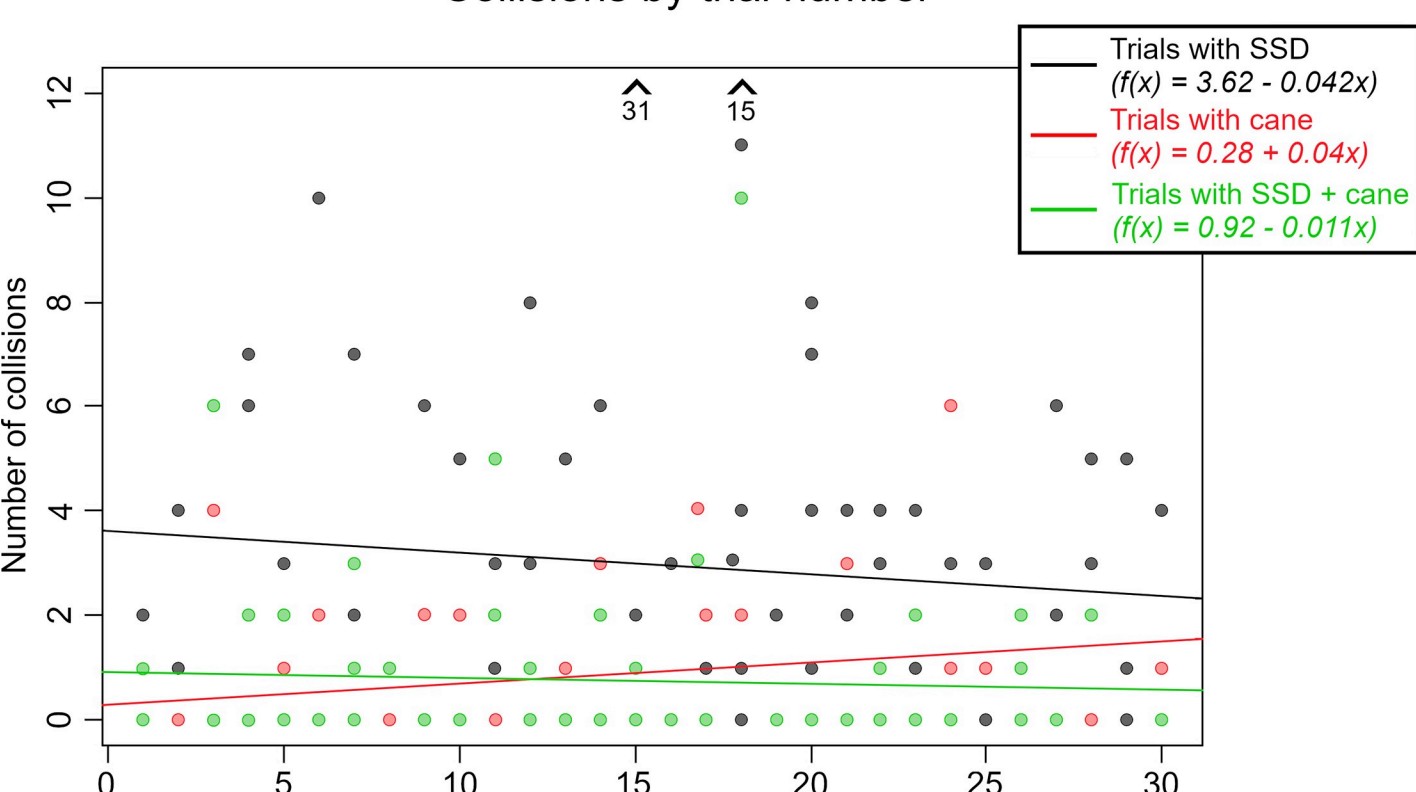

**Fig 5. Collisions in navigation trial.** Collisions in relation to the trial number as well as the regression lines for each condition. The two arrows indicate two data points for SSD trials which surpass the scale of the y-axis.

The first question of the evaluation showed that 83% of the participants subjectively rate their own learning progress positively, with an average of 3.75 on a scale from 1 = 'no learning progress' to 5 = 'very high learning progress'.

Following the results of the second question, four participants would also expect to have further progressed at a similar rate if the training would have continued, whereas one participant expected a slower learning progress and one participant expected a faster learning progress.

The third question shows that 100% of the participants saw high potential for the application of the designed SSD in obstacle awareness. 67% could imagine using it in living spaces, 50% for navigation in public buildings. 33% saw potential for the device in traffic-free zones. None of the participants saw potential for the device for navigation in traffic or for object recognition.

The fourth question of the evaluation covered the advantages and disadvantages of the SSD. Each participant gave individual answers, which are summarized by topic. 100% noted the impractical and bulky design of the device. 67% noted the advantage of increased range to perceive objects and obstacles. 67% noted the disadvantage of the auditory output of the SSD interfering with real-world sounds. 50% mentioned the advantage of detecting gaps in a wall or line of obstacles. One person mentioned the difficulty to precisely determine the distance

**Table 2. Object recognition results.**

| Scene | Answer 1 | Answer 2 | Answer 3 | Answer 4 | % correct | Avg. time |
|-------|----------|----------|----------|----------|-----------|-----------|
| 1 | **Table** | Car | Garden fence | Wardrobe | 71.43 | 140s |
| 2 | Wardrobe | Car | **Staircase** | Door | 100 | 65s |
| 3 | Bus stop sign | Trash can | Window | **Tree** | 57.14 | 71s |
| 4 | Trash can | **Chair** | Nightstand | Bathtub | 28.57 | 68s |
| 5 | Table | Bathtub | **Car** | Trash can | 28.57 | 104s |
| 6 | Staircase | Window | Table | **Door** | 42.86 | 103s |
| 7 | **Bus stop sign** | Bollard | Door | Human person | 85.71 | 89s |
| 8 | Couch | Staircase | **Window** | Bed | 33.33 | 94s |
| 9 | Chair | Bollard | **Trash can** | Bathtub | 28.57 | 74s |
| 10 | Car | **Couch** | Bathtub | Table | 42.86 | 107s |
| 11 | Trash can | Bollard | **Chair** | Guitar | 42.86 | 90s |
| 12 | **Car (in front of wall)** | Table (in front of wall) | Staircase | Door | 57.14 | 103s |
| 13 | Bus stop sign | Window | Human person | **Tree** | 85.71 | 57s |
| 14 | Bus stop shelter and sign | Couch and coffee table | **Table and chairs** | Park bench and trash can | 42.86 | 142s |
| 15 | Group of people | **Park bench and trash can** | Multiple trees | Bus stop shelter and sign | 50 | 155s |
| 16 | Table | Staircase | Bus stop shelter | **Couch** | 50 | 106s |

A list of all sets of answers, with the presented object marked bold, as well as the percentage of correct identifications and the average time required to identify the object. The chance at even distribution is 25%.

based on the auditory volume. One person mentioned the disadvantage of a long training duration.

The fifth question asked for desired adjustments or optimizations of the device. 33% suggested to use bone conducting headphones to free the ear canals. 33% suggested to use haptic feedback instead of auditory feedback. One person suggested to translate color or surface structure into some form of auditory stimulus.

## Discussion

The results demonstrate the feasibility and intuitive usability of the designed sensory substitution algorithm: participants were able to use it to navigate through complex obstacle courses and correctly identify 3D objects within the very first trials. In addition, the technological feasibility to use a consumer-level smartphone for spatial-information-based sensory substitution was confirmed. However, within ten hours of training, the SSD could not outperform the white cane as navigational aid. This finding is expected for SSD-only trials, as their main purpose in the study was to train and familiarize participants with the new type of navigation aid without the option to ignore it and rely solely on the white cane. It is also consistent with previous studies of similar devices [37]. The reason is that all participants already had years of training with the white cane, whereas the SSD poses as a new and unfamiliar experience. However, the hypothesis of the SSD being able to improve the overall navigation performance of participants when combined with a white cane could also not be shown within the duration of training. This is shown by the navigation performance with SSD+cane not significantly varying from those with just the cane, both in average collisions and required navigation time. While this may be, as will be later discussed, only a result of lack of training rather than a showcase of the absolute limit of the device, it certainly indicates that the device cannot be seen as instantaneous and learning-free improvement to standard navigation with a white cane.

Ideally for such an experiment, the recruited blind participants would have no experience with the white cane. This way, they would be introduced to both navigational aids at the same time. Only in this scenario results would be comparable. However, in a practical setup such a limiting participation criterion is not feasible, as about 80% of blind or legally blind persons have experience with a white cane [3]. With blindfolded sighted participants, this lack of participants not familiar with the white cane could be tackled. However, multiple studies found that results of blindfolded participant performance in navigation and recognition tasks are not representative of the performance of blind participants [15, 48, 49].

It must also be noted that the effective training time–meaning the time in which the participant actively used the SSD–is considerably lower than the total duration of the study of ten hours. Considering the navigation trials in which only the white cane was used and the time that was required in between each navigation trial to change the obstacle course, the real training duration sums up to around one hour of tutorial and two to three hours of navigation and obstacle recognition. As a consequence, even at the end of the study, participants have used the SSD only for around four hours. The tendencies found in the regression lines of both the required time and number of collisions in the navigation task, as well as the participants' positive feedback, support the hypothesis that with extensive training, a significant performance increase could be shown, and that four hours of training do not suffice to achieve significant learning effects.

The correct identification rate in the object recognition trials was significantly above chance level even without any previous training and without previously showing the objects and structures to the participants. This is an important finding, as it shows that even participants who have been blind since birth or childhood interpreted the auditory output of the SSD well enough to recognize objects by their visual shapes.

However, the concluding evaluation revealed that none of the participants saw potential for the developed translation algorithm in the task of object recognition. This contrasts with the positive recognition rate observed in the experiment. The most likely explanation for this contradiction is the slow identification rate of almost 100 seconds per object on average, as it makes this field of application impractical for everyday life. These findings suggest a follow-up study with stronger focus on object recognition using the EASV, including training phases and feedback questions aimed more towards this function and its usefulness in everyday life. Such a study allows to assess potential reasons for the difference between objective results and subjective rating of object recognition and whether the occurrence of this difference is limited to the first trials or persists even through longer training periods.

Direct comparisons of the results found for the performance of the SSD in this study and the results of previous studies are not possible due to the differences in experimental conditions. Conditions differ e.g. by the type of environment–virtual, augmented or real–, the layout of the obstacle courses or the method to measure the correct identification of objects. However, a qualitative comparison to other studies provides new insights. Specifically, it provides a better overview of the strengths and weaknesses of the EASV and allows assumptions about how the performance would develop over longer training durations.

In a study conducted by Malika Auvray, Sylvain Hanneton and John Kevin O'Regan [29], the ability of sighted blindfolded people to recognize common objects, such as a table, bottle or book, using 'the vOICe' was tested. It was found that over the course of 50 randomized trials per participant, the time until the object was correctly recognized decreased by around 40% over the course of the study. A similar result was found in another study using the same device, where again a significant decrease in the time required to identify the object was shown over the course of 36 trials [30]. This indicates that object recognition tasks with an SSD can greatly enhance recognition speed through training, implying that the long time needed by

participants to identify objects with the EASV is likely caused by insufficient training. It can be assumed that some objects are statistically more difficult to identify due to their shape or complexity, which will also have an influence on the comparability of the studies. However, an objective assessment of that difficulty is a topic that—although already adressed in different research papers, such as the ones by Heung-Yeung Shum [50] or Longin Jan Latecki and Rolf Lakämper [51]—goes beyond the scope of the analysis in this study.

In a navigation study with both real and virtual maze environments, the navigation performance of the participants using the EyeCane was found to significantly improve, with both the average required time to navigate through the maze and the average collisions per trial being reduced to around half after 3 sessions of 90 to 120 minute training [52]. However, in the group of low vision and late blind participants, which is most comparable to the study population of the EASV study, the time required to solve the maze was 152s on average. That is significantly higher than the average required navigation time measured for the SSD-only trials with the EASV of 68±55s, despite an overall shorter distance from start to finish. While these differences in time may originate simply from differences of the experimental setup, such as the complexity of the obstacle course, it further suggests a low entry barrier of the EASV.

In a study using the Sound of Vision SSD, a significant reduction of collisions was detected within the first five trials [37]. The reduction was achieved by two hours of training in between each trial and additionally four hours of introductory training in a virtual and real-world environment each. Given that no training was done in between the trials of the EASV study and the overall shorter training duration, the results of the study by Hoffmann et al. indicate that a significant decrease in collision rate might be found for the EASV device within ten to twelve hours of effective training. However, the study by Hoffmann et al. did not show a significant decrease in the required time per trial and they also concluded that a much longer training duration is required for the navigation speed to significantly increase.

The participant feedback on the SSD that was gathered in the last two questions of the evaluation revealed various potential optimizations to the translation algorithm as well as aspects that should be considered in the future development of SSDs using such algorithms. First, the overall hardware of the device must be changed in order to be more comfortable, subtle and less obtrusive and also in order to no longer cover up the visual field, allowing to fully use any remaining vision. This is especially important when considering the development of a prototype for use in everyday life. The lack of subtlety and wearing comfort is a problem that many SSDs have not yet overcome and that is seen as a major factor that still prevents blind persons from using SSDs [20, 53, 54]. A possible solution is to use a separate input device as opposed to the internal smartphone camera, where only the input device is head-mounted, for example in form of a small wireless camera attached to spectacle frames. Since the EASV only requires a single image sensor for visual input, this solution is more subtle than in a setup with multiple image- and depth sensors.

To oppose the interference of output signals with real-world sounds, the SSD should deploy bone conduction. While in this case, the output signals of the SSD still demand mental capacity of the auditory cortex, the auditory canals are unblocked and thus, all sounds from the real world can be perceived unimpeded. One participant mentioned that the distance of objects is not easily detectable, since it is indicated purely by volume and there is no intrinsic factor between volume and distance. It is likely that with more training, users would achieve a better understanding of the distance based on volume, but in order to make the algorithm more intuitive, additional auditory stimuli should be considered for the translation of distance. Further, to support especially the object recognition function of the device, the color or surface structure captured by the center field of the visual input image could be translated into different variations of non-monotone sounds. A similar method of color translation, using different

instrumental sounds, was explored and successfully applied for object recognition tasks with the EyeMusic SSD [32]. However, such an implementation of additional types of auditory stimuli must be carefully weighed up against the increased information load on the auditory sense.

It could further be argued that the visual vertical position should translate into auditory vertical position instead of pitch (see Fig 1(A) for the individual translations of stimuli). The decision to use pitch was made in regard to the fact that the auditory perception of vertical position purely depends on the head related transfer functions [55]. This would require the test person to resolve the slight changes in frequency and amplitude a sound wave experiences when being reflected by the head and auricle. Because of this, the accuracy in determining auditory vertical position is very low in comparison to other auditory stimuli [25].

While the EASV is not yet capable of capturing real-world obstacles without preliminary marking and modeling of the obstacles, it is already foreseeable that this will be possible in the near future. ARCore recently announced a new feature which allows to calculate depth- and surface maps in real-time using a smartphone [56]. If this technology is fast and accurate enough to be viable for navigation, the software of the EASV could be easily adjusted to allow for any modern smartphone to be used as a spatial-information-based visual-to-auditory SSD. In combination with the proposed optimizations especially to the hardware design, the EASV would offer an easily accessible and intuitive introduction to sensory substitution, which would likely increase the acceptance of SSDs as additional navigational vision aids.

## Conclusion

In this study, the feasibility and performance of a spatial-information-based SSD was evaluated in a navigational task, using a simulated environment synchronized with real-world objects. The performance of the SSD was benchmarked against the performance with a white cane and with a combination of both SSD and white cane. In addition, the ability to correctly identify 3D objects and structures using the same device was assessed by presenting virtual objects and a selection of four answers. The participants' feedback and perception of their learning progress was assessed in a concluding evaluation.

It was found that all participants were able to successfully navigate using the SSD, even immediately after the instructions. Without any prior training in object recognition and only three sessions using the SSD, participants recognized complex 3D objects such as cars, chairs and staircases when presented in otherwise empty environments. Further, this was done using a consumer-level smartphone, showing the technological feasibility of depth-based sensory substitution without the use of expensive peripheral devices.

However, the study also reveals the limitations of the developed SSD. Within the duration of the training, the SSD could offer no significant advantage compared to navigation with the white cane, even when both these navigational aids were used in combination. Further, while object recognition did show a significant success rate, the average time of almost 100 seconds that was required to recognize one object makes this feature impractical for real-world application without more extensive training.

## Supporting information

**S1 Fig. Study timetable.** The timetable for the five sessions of the study.
(TIF)

**S2 Fig. Obstacle courses.** The 23 layouts of the obstacle courses used for the navigation task.
(TIF)

**S1 File. Navigation results.** The raw data of the navigation trial results.
(XLSX)

**S2 File. Object recognition results.** The raw data of the object recognition trial results.
(XLSX)

**S3 File. Evaluation results.** The results of the concluding evaluation.
(XLSX)

**S1 Video. Navigation showcase.** Showcase video of a navigation trial using the developed SSD.
(MP4)

# Acknowledgments

We thank Professor Frank Bremmer of the Research Group Neurophysics of the Philipps-University Marburg for his patronage on the study and Mrs. Denise Peter of the university's room reservation management. Further, we want to thank the participants of this study and all people who helped in the design and development of the device through testing and feedback.

# Author Contributions

**Conceptualization:** Alexander Neugebauer, Katharina Rifai, Siegfried Wahl.

**Data curation:** Alexander Neugebauer.

**Formal analysis:** Alexander Neugebauer.

**Funding acquisition:** Siegfried Wahl.

**Methodology:** Alexander Neugebauer.

**Project administration:** Katharina Rifai.

**Software:** Alexander Neugebauer.

**Supervision:** Katharina Rifai, Mathias Getzlaff, Siegfried Wahl.

**Visualization:** Alexander Neugebauer.

**Writing – original draft:** Alexander Neugebauer.

**Writing – review & editing:** Katharina Rifai, Siegfried Wahl.

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
