## [Decision Letter · Decision Letter 0]

5 May 2020

PONE-D-20-09308

Navigation Aid for Blind Persons by Augmented-Reality based Visual-to-Auditory Sensory Substitution

PLOS ONE

Dear M. Neugebauer,

Thank you for submitting your manuscript to PLOS ONE. After careful consideration, we feel that it has merit but does not fully meet PLOS ONE’s publication criteria as it currently stands. Therefore, we invite you to submit a revised version of the manuscript that addresses the points raised during the review process. Both reviewers had many comments concerning the knowledge of the literature in the field and some important methodological issues.

Your revised manuscript will be re-submitted to the reviewers.

We would appreciate receiving your revised manuscript by Jun 19 2020 11:59PM. To enhance the reproducibility of your results, we recommend that if applicable you deposit your laboratory protocols in protocols.io, where a protocol can be assigned its own identifier (DOI) such that it can be cited independently in the future. For instructions see: http://journals.plos.org/plosone/s/submission-guidelines#loc-laboratory-protocols

We look forward to receiving your revised manuscript.

Kind regards,

Maurice Ptito

Academic Editor

PLOS ONE

We note that one or more of the authors are employed by a commercial company: Carl Zeiss Vision International GmbH

"Funding received from University of Tuebingen (ZUK 63) as part of the German Excellence initiative from the Federal Ministry of Education and Research – Germany (BMBF). This work was done in an industry-on-campus-cooperation between the University of Tuebingen and Carl Zeiss Vision International GmbH."

"The authors received no specific funding for this work."

Additionally, because some of your funding information pertains to commercial funding, we ask you to provide an updated Competing Interests statement, declaring all sources of commercial funding.

In your Competing Interests statement, please confirm that your commercial funding does not alter your adherence to PLOS ONE Editorial policies and criteria by including the following statement: "This does not alter our adherence to PLOS ONE policies on sharing data and materials.” as detailed online in our guide for authors  http://journals.plos.org/plosone/s/competing-interests.  If this statement is not true and your adherence to PLOS policies on sharing data and materials is altered, please explain how.

Please include the updated Competing Interests Statement and Funding Statement in your cover letter. We will change the online submission form on your behalf.

4. We note you have included a table to which you do not refer in the text of your manuscript. Please ensure that you refer to Table 1 in your text; if accepted, production will need this reference to link the reader to the Table.

Reviewers' comments:

Reviewer's Responses to Questions

**Comments to the Author**

1. Is the manuscript technically sound, and do the data support the conclusions?

Reviewer #1: Partly

Reviewer #2: No

2. Has the statistical analysis been performed appropriately and rigorously? 

Reviewer #1: Yes

Reviewer #2: No

3. Have the authors made all data underlying the findings in their manuscript fully available?

Reviewer #1: Yes

Reviewer #2: Yes

4. Is the manuscript presented in an intelligible fashion and written in standard English?

Reviewer #1: No

Reviewer #2: Yes

5. Review Comments to the Author

Reviewer #1: Thank you for giving me the opportunity to review this manuscript. The authors present a study using a sensory substitution apprioach to the detection and identification of obstracles during mobility by persons living with vision impairment. The comments below are intended to further support the authors to maximize the presentation of their outcomes.

Abstract, add the actual number of participants

Line 46, replace “suffer from” with “living with” as these people may not all be suffering.

Line 47 and throughout manuscript, replace “impact” with “effect”, as impact implies the collision of one object on another (general use in physics).

Line 48 “only few advances” this is a very general statement, please provide citation, otherwise this is too subjective.

Line 50 – restoration implies that a previously lost function is restored. The authors likely want to refer to rehabilitation as an approach to introduce strategies and tools toimprove an ability that is recovered in an alternative format. Depending on the subspecialty of research (low vision rehab versus engineering), the audience (e.g., rehabilitation professionals) and the original language (English, French and German do not use equally translatable terms), there are differences in terminology. The authors may need to define their terms to avoid confusion. Sensory substitution is one approach to the rehabilitation of vision impairment, but has nothing to do with restoration, a term I would accept in the context of reconstructing the retinal for example.

Line 50 to 54, this description requires more detailed examples and concrete terms. Right now this is at the very surface, and given the importance of this concept, needs to be clarified in more detail.

Line 55 needs slight grammatical correction because sensory substitution is an approach, not another type of visual aid.

Line 58, replace “real” sense with “original” sense.

Line 63, should be recognised “to contain the equivalent of visual information”. The authors need to generally be more careful with their wording as many phrases lack the subtlety to express what they are after.

Line 67, many researchers in speech perception would beg to differ when it comes to the ability to process complex signals in hearing. The authors need to refine their statement here.

Line 87, please correct the citation.

Line 140, replace “aging from” with “aged”; why “to over 60”? Please provide details on age, mean, range, in table 1 with additional participant characteristics. Diagnosis to cause vision loss etc. to inform a clinical reader. The degree of blindness column makes no sense, this is not how vison loss is expressed. We need actual visual acuities, ideally for each eye, and binocularly, as well as visual field measures, for each eye and binocularly. If this info is not available, the authors really need to justify why, as this is essential for the credibility of their participants, and would account for many aspects of their performance. Is the visual acuity expressed in decimal units? Please clarify.

Line 152, we need more details on the audiometric measures here, besides the up/down method. Which frequencies were measured, what were the individual thresholds for each ear (add this in table 1), under what conditions was the measurement made, in a sound booth? Any efforts to control the influence of noise? More detail is better to convince the reader that hearing was normal or suitable for the task. What were the audiometric characteristics of the stimuli to determine whether people could hear the stimuli…

Line 194, why the 7 minutes?

Line 285-290, these response are better to be reported in the text as percentages here and the corresponding figures 11, 12 and 13 can be deleted.

Line 298, qualitative results should be reported in the Results section and discussed in the discussion. Please insert some type of reporting here, e.g. a table of relevant quotes, a list of relevant themes.

Given that there are qualitative aspects to the study, this needs to be mentioned in the methodology and not just appear out of nowhere in the results section. A citation or two on the qualitative analysis approach to the data are needed.

Line 313 onwards “within ten hours of training, the SSD could not outperform the white cane as navigational aid. This finding is expected, as it is consistent with previous studies.” This statement puts in question why you would spend all this time and effort developing something when you do not expect that it will improve on the current clinical standard of care – the while cane. The authors need to frame this fact in order to explain better why the study was conducted in the first place.

In comparison, the object identification data demonstrate the potential use of the device, as that would go above and beyond what a cane can do. The authors need to maximize this finding. – Interesting that the participant data did not support this result, as the participants immediately realized that this practical implication is not all that practical. So there is some work to be done to improve this potential?

Line 372 – “Thanks to the close collaboration with blind persons” would indicate that this was a participatory action research? Did the authors actually consult with blind persons BEFORE conducting the study, or were they involved in the design of its priorities, questions to be asked, etc.? Were they involved in the analysis and interpretation of the data or the preparation of this manuscript. These aspects would make it a close collaboration. Please clarify.

Mention in limitations: The full-field virtual set-up eliminates the use of any remaining visual field. Is this a worth price paying for what is gained by the device?

Reviewer #2: Dear Authors,

I read your interesting study investigating the usability of a novel SSD developed by your team to improve the performance of blind participants in navigation and object recognition. It is an interesting study with a lot of merit but there are some very important conceptual and methodological flaws in the study, as well as an overall lack of the knowledge on the literature in this field. I found that your conclusions were interesting, but not supported by the data. This is a shame! There is a real need for the development of a usable, affordable SSD that could be based on an algorithm using existing technology.

MAJOR ISSUES:

1. There are only 7 participants in the study. Even for a study comparing completely blind, congenitally blind participants, this is a very small sample. According to the description of your participants given in the methods section however, most of your participants would be considered low vision (or late blind) according to the definition that is usually given in the literature. Congenitally blind (CB) participants with no light perception are indeed very hard to find, and sometimes a sample of only 10 CB can be acceptable. In the case of low vision participants, it is somewhat relatively easier to find participants, thus I really think this sample of only 7 participants is inadequate. Can you possibly increase the sample size for this group to 10 participants minimum?

2. The lack of control group in the study. As I mention in my previous comment, there are only 7 participants in the whole study... and no control group. A control group of blindfolded sighted individuals could have been added to this study making teh results much more interesting. I understand that participants served as their own controls in the study, the white cane condition vs. the SSD condition vs. the combined condition, but this comparison only makes sense if you compare it with a group of participants with no experience with the white cane, as you correctly mention in your discussion. You could have disentangled the issue of experience with the white cane. Furthermore, it is generally accepted that CB participants who are completely blind with no light perception benefit from mechanisms of brain plasticity that enable them to recruit visual brain areas for tasks involving other modalities (for example via SSD). This usually affords them better behavioral performance with these devices. Thus, it is unclear if CB participants could have performed better than this group of low vision participants in the current study.

3.Participants get better at a task each time they do it. Your study compared participants with SSD vs. the same participants with SSD and cane, the same participants again with only a white cane. It is unclear what measures were taken to circumvent the repetition effect. Did participants do the tasks in different orders, enabling you to compare performances between those subjects who did the cane first part and then the SSD part and the combined part? This is unlikely given the small sample size! If no measures were taken, or worst, if all participants did the SSD part first and the cane part of the study, your effects might simply be due to this repetition effect. Could you please explain what was done to control for this?

4. Lack of knowledge of the literature. Although you do cite some key studies in your Introduction, you do not cite THE key studies that would be relevant to this study. Experiments on object recognition and spatial navigation, reviews on brain plasticity and SSD use, reviews offering explanations why SSDs are not commonplace. You also sometimes cite teh wrong studies, for example you cite teh concept of brain plasticity and offer a citation, but this study is not the one that coined the term brain plasticity. Also, there have been several studies using SSDs for obstacle avoidance and navigation, and these studies are not discussed. Other studies have shown that it is possible to use a SSD to navigate as efficiently as sighted participants using vision in simple environments, or that in certain situations, SSDs can give you information that enables to navigate more efficiently than a white cane. There are even studies that have used virtual reality to train participants, and that transfer between real and virtual environments was possible for participants of varying degrees of visual expé rience, which could be an interesting point for the discussion of yoru results. The fact that you omit these important papers gives the impression that you are unaware of these studies.

5. Conclusions. Your results simply do not support your conclusions! You say some interesting this ion the discussion section about the future of SSDs , and where to place the camera, etc, but nothing in yoru results suggest these conclusions, even remotely. The Introduction and Discussion would be better by focusing on the themes of this study: Navigation and object recognition with a novel SSD.

6. Learning rate. The main thing that you claim to be testing in this study is learning rate. It is unclear how your compared performances to asses if indeed learning had taken place. Usually comparisons between early training and late training is assessed, or comparisons between novies and experts. Is the only way that you assessed training by self-report?

MINOR ISSUES:

1.Figures. Need to be prepared with Illustrator or photoshop. The copied images into the text look grainy and not very clear (ex. Fig 1, Fig 6, 7, 11, 12, &13). There is no reason for images with text to be grainy. There are alot of figures. Maybe you could put methods figures in one ,a nd the results figures together? Might I suggest making one single figure or possibly even only 2 for the methods ? Making one figure from figures 1-6 could really help clarify the information and make it more succinct. The results figures need some work also and are very grainy (low resolution) and not very clear. Maybe a bar graph would be more adequate?

2. Methods. Not enough information is given about teh device itself, its field of view and method of translating visual information. Maybe a video file uploaded as supplementary material would be helpful? Also, it might be useful, at least for the sake of this paper, to give your SSD a name ! The Central Sound View? Because it translates the central visual field into sounds?

3. It is not clear from your methods section how object recognition was done. Were they simply recognizing the obstacles in the obstacle course?

6. PLOS authors have the option to publish the peer review history of their article (what does this mean?). If published, this will include your full peer review and any attached files.

Reviewer #1: No

Reviewer #2: No

---

## [Author Response · Author response to Decision Letter 0]

23 Jun 2020

Dear Reviewers,

Thank you very much for taking the time to read our manuscript and for the valuable feedback. All points gave us great advice to further improve the paper, although we couldn't address all issues raised by the reviewers, partially due to the Covid-19 situation. For example, it will not be possible to increase the sample size or recruit participants for a control group soon. However, we did reach out to the participants of the study, asking to voluntarily share information about their age and diagnostic background of their visual impairment. We think that the manuscript, after addressing the issues raised by the reviewers, is a valuable contribution to the field and we hope that you can consider it for publication.

In response to the first review:

Comment: Abstract, add the actual number of participants

Response: Done.

Comment: Line 46, replace "suffer from" with "living with".

Response: Done.

Comment: Line 47 and throughout manuscript, replace "impact" with "effect", as impact implies the collision of one object on another (general use in physics). 

Response: Done.

Comment: Line 50 - restoration implies that a previously lost function is restored. The authors likely want to refer to rehabilitation as an approach to introduce strategies and tools to improve an ability that is recovered in an alternative format. Depending on the subspecialty of research (low vision rehab versus engineering), the audience (e.g., rehabilitation professionals) and the original language (English, French and German do not use equally translatable terms), there are differences in terminology. The authors may need to define their terms to avoid confusion. Sensory substitution is one approach to the rehabilitation of vision impairment, but has nothing to do with restoration, a term I would accept in the context of reconstructing the retinal for example.

Response: Good point. Changed to „rehabilitation“. 

Comment: Line 58 replace "real" sense with "original" sense

Response: Done.

Comment: Line 87 please correct the citation

Response: Done.

Comment: Line 140 replace "aging from" with "aged"

Response: Done.

Comment: Line 48 "only few advances" this is a very general statement, please provide citation, otherwise this is too subjective.

Response: Thank you. As was already correctly pointed out, the term "vision restoration" should be and was replaced with "vision rehabilitation". Further, to specify the focus on navigation the phrase "[…] only few advances in the field of vision restoration and assistance tools for the blind have reached widespread use in public within the last decades " was changed. See below the change. Several citations were added to support this statement.

Change in the manuscript (lines 47-49): 

• "[…]only few advances in the field of vision rehabilitation for navigation and navigation assistance tools for the blind have reached widespread use in public within the last decades, with the white cane still being the most common assistance tool for the blind [3-5]."

Publications added:

• R. Manduchi, S. Kurniawan - 2011 - Mobility-Related Accidents Experienced by People with Visual Impairment

• J. Rizzo – 2019 - A new primary mobility tool for the visually impaired: A white cane—adaptive mobility device hybrid

• R. Jafri – 2018 - User-centered design of a depth data based obstacle detection and avoidance system for the visually impaired

Comment: Line 50 to 54, this description requires more detailed examples and concrete terms. Right now this is at the very surface, and given the importance of this concept, needs to be clarified in more detail.

Response: We appreciate the comment. A more detailed description of the structure and working process of a sensory substitution device, as well as a more general description of sensory substitution was added. 

Change in the manuscript (lines 51-62):

• “The term was first introduced in 1969 by the neuroscientist Paul Bach-y-Rita [6] and describes a non-invasive process to translate sensory stimuli like brightness or color into stimuli of a different sense, for example volume and pitch. It is most often used to compensate for the loss of one sense and works in all cases where the neuronal function of the brain is unaffected by the damage causing this disabily [7,8]. The sensory information is translated by a sensory substitution device (SSD), which typically consists of three parts: The input device that captures information of the type to be translated, a processing unit that translates the information following a specified algorithm, and an output device that transmits the translated information to the user. As an example, a visual-to-auditory SSD would require a visual input device such as a camera, a processing unit like a smartphone or notebook, and an auditory output device such as headphones.”

Publications added:

• P. Bach-y-Rita – 1969 – Vision substitution by tactile image projection

Comment: Line 55 needs slight grammatical correction because sensory substitution is an approach, not another type of visual aid

Response: The phrase "What sensory substitution excels at […]" was changed to "What sensory-substitution-based devices excel at […]" (line 63 in the revised manuscript)

Comment: Line 63, should be recognised "to contain the equivalent of visual information". The authors need to generally be more careful with their wording as many phrases lack the subtlety to express what they are after.

Response: This sentence was indeed misleading, as it may suggest that the output signal would be a visual signal, which is not the case. The sentence "This indicates that the output signal is recognized as visual information even though it is an auditory or tactile stimulus." was changed as follows.

Changes in the manuscript (lines 72-74)

• "This indicates that the output signal, despite being an auditory or tactile stimulus, is perceived and processed in the same way as a visual stimulus."

Comment: Line 67, many researchers in speech perception would beg to differ when it comes to the ability to process complex signals in hearing. The authors need to refine their statement here.

Response: Thanks for the comment. The text passage "However, the capacity to process information varies greatly between the different senses with the visual sense being able to process manifolds the amount of information of every other sense. " was changed as follows. 

Changes in the manuscript (lines 75 – 78): 

• "However, the capacity to process information was found to vary greatly between the different senses, with the visual sense being able to process manifolds the amount of information of every other sense [1, 17, 18], as can be estimated by the number of fibers and their spike rates in optic, auditory or haptic nerves [19-23]."

Publications added:

• P. Ash – 1951 – The sensory capacities of infrahuman mammals: vision, audition, gustation

• W. Bialek – 1993 – Bits and brains: Information flow in the nervous system

• K. Kokjer – 1987 - The information capacity of the human fingertip

• Kristjansson – 2016 - Designing sensory-substitution devices: Principles, pitfalls and potential

• M. Proulx – 2014 - Multisensory perceptual learning and sensory substitution

• R. Wurtz – 2000 – Central Visual Pathways

Comment: Line 140(A) - why "to over 60"? Please provide details on age, mean, range, in table 1 with additional participant characteristics. Diagnosis to cause vision loss etc. to inform a clinical reader.

Response: Thanks for bringing this up. The reason for this is that we prioritized the comfort of the participants by minimizing the requirement to share personal information in the questionnaire. We understand that this procedure is not ideal, and have reached out to our participants, asking them to voluntarily share the missing information. The information was added in table 1.

Comment: Line 140(B) The degree of blindness column makes no sense, this is not how vison loss is expressed. We need actual visual acuities, ideally for each eye, and binocularly, as well as visual field measures, for each eye and binocularly. If this info is not available, the authors really need to justify why, as this is essential for the credibility of their participants, and would account for many aspects of their performance. Is the visual acuity expressed in decimal units? Please clarify.

Response: Thanks for pointing to this. The information about visual acuity was adjusted to the category of visual impairment, based on the ICD-10 Codes for Blindness (H54). The table also includes the clinical picture of the participants. This information is according to the statement of the participants, as we did not have the required devices to measure the visual acuity or visual field of the participants in the city in which the study was carried out. (Due to the high percentage of blind persons among the residents, the study was carried out in Marburg. This will be explained in more detail in point 1 of the 'major issues' section of the second review. All authors' affiliations, meanwhile, are either located in Tübingen or Düsseldorf, thus we had no access to the required medical devices in Marburg.)

However, while it is true that the visual acuity and visual field influence the navigation of the participants in everyday life, the participants were wearing the SSD during all trials of the experiment – even in trials where only the white cane was used for navigation, as the SSD was still used to track the time and collisions during the trial. Since the current hardware setup of the SSD completely covers the visual field, it was not possible for the participants to use their remaining eyesight during the experiments. This information was missing in the original version of the manuscript and was now added.

It can be assumed that people with relatively high remaining eyesight will use the white cane differently to those who must completely rely on the white cane and their other senses, which changes their overall performance with the cane. The difference in performance between participants, however, is already considered as a random factor of the linear mixed model in the analysis of the results. 

Since the direct influence of remaining eyesight in the navigation experiment can be precluded and the indirect influence – in form of different navigation behaviour and white cane expertise – is considered in the analysis, we think that the results of this study are still meaningful.

Changes in the manuscript (lines 248-251)

• “The hardware setup of the SSD was worn during all navigation trials to measure time and collisions. Further, this method ensured that the results are not influenced by the remaining eyesight of the participants. In trials in which only the cane was used for navigation, the auditory output of the SSD was muted.”

Comment: Line 152 "we need more details on the audiometric measures here, besides the up/down method. Which frequencies were measured, what were the individual thresholds for each ear (add this in table 1), under what conditions was the measurement made, in a sound booth? Any efforts to control the influence of noise? More detail is better to convince the reader that hearing was normal or suitable for the task. What were the audiometric characteristics of the stimuli to determine whether people could hear the stimuli…"

Response: Good point. Originally, we did not want to put too much focus on the audiometry, as it was not a medically meaningful test and the hearing performance was not measured in decibel, but instead only in reference to the minimum volume threshold of the SSD's auditory output. However, we agree that more information about the procedure of the audiometry will better convey the meaning of the results. The results of the audiometry were added to table 1, as suggested.

Changes in the manuscript (lines 190-204):

• "The audiometry used 0.5 second sound samples of five different frequencies (143Hz, 220Hz, 440Hz, 593Jz, 880Hz) covering the frequency range of the auditory output of the SSD. Each frequency was tested for 60 seconds, with randomized 4 to 10 second intervals between individual sound cues. After each correct response to the sound cue, the volume of the next cue was decreased by 33%, for each missed or false response, the volume was increased by 50%. At 60 seconds, the volume of the last recognized sound cue was saved, and the test automatically proceeded with the next frequency. All volumes were measured in software-internal, unitless values, allowing the direct comparison to the auditory output of the SSD which uses the same software-internal values with a minimum volume threshold of 0.1. The results of the audiometry are listed in Tab 1, with all results below the threshold of 0.1 indicating that the participant is able to fully perceive the auditory output in the respective frequency.

The audiometry was done in the same room and under the same noise conditions as the main experiments. There was no test for each individual ear as the auditory output of the SSD is monaural."

Comment: Line 194, why the 7 minutes?

Response: The duration of seven minutes per trial was estimated based on two factors: 

1. the time required for moving the obstacles to their new required position, which was tested before the experiments and found to level out at around 4 minutes, varying between maps.

2. the time for the navigation. As it was not certain how long it would take the participants on average to navigate through the obstacle course, we assumed a rather long time of 2 minutes on average. As we found in the results, this was more than twice the actual average required time, but we preferred having some time buffer. 

Another minute was included in the calculation for questions/remarks from the participants or other conversation, adding up to a total of 7 minutes. The text in line 245 of the revised manuscript was slightly adjusted to now specifically mention the time required to restructure the obstacle course: "For each navigation trial a total duration of 7 minutes was estimated, including 4 minutes to restructure the obstacle course in-between trials."

Comment: Lines 285-290, these response are better to be reported in the text as percentages here and the corresponding figures 11, 12 and 13 can be deleted.

Response: We changed the respective text passage in the revised manuscript version and removed the figures as recommended.

Changes in the manuscript (lines 343-365):

• “The first question of the evaluation showed that 83% of the participants subjectively rate their own learning progress positively, with an average of 3.75 on a scale from 1 = ‘no learning progress’ to 5 = ‘very high learning progress’. 

Following the results of the second question, four participants would also expect to have further progressed at a similar rate if the training would have continued, whereas one participant expected a slower learning progress and one participant expected a faster learning progress. 

The third question shows that 100% of the participants saw high potential for the application of the designed SSD in obstacle awareness. 67% could imagine using it in living spaces, 50% for navigation in public buildings. 33% saw potential for the device in traffic-free zones. None of the participants saw potential for the device for navigation in traffic or for object recognition.

The fourth question of the evaluation covered the advantages and disadvantages of the SSD. Each participant gave individual answers, which are summarized by topic. 100% noted the impractical and bulky design of the device. 67% noted the advantage of increased range to perceive objects and obstacles. 67% noted the disadvantage of the auditory output of the SSD interfering with real-world sounds. 50% mentioned the advantage of detecting gaps in a wall or line of obstacles. One person mentioned the difficulty to precisely determine the distance based on the auditory volume. One person mentioned the disadvantage of a long training duration.

The fifth question asked for desired adjustments or optimizations of the device. 33% suggested to use bone conducting headphones to free the ear canals. 33% suggested to use haptic feedback instead of auditory feedback. One person suggested to translate color or surface structure into some form of auditory stimulus.”

Comment: Line 298, qualitative results should be reported in the Results section and discussed in the discussion. Please insert some type of reporting here, e.g. a table of relevant quotes, a list of relevant themes. Given that there are qualitative aspects to the study, this needs to be mentioned in the methodology and not just appear out of nowhere in the results section. A citation or two on the qualitative analysis approach to the data are needed.

Response: Thanks for the comment. We added more information on the concluding evaluation in the methodology part, including a citation.

Further, the text passage in the statistical methods section, starting at line 300, was shortened, as this information overlapped with the new text passage in the methodology.

Changes in the manuscript (lines 267-272):

• “A concluding evaluation, comprised of five questions, assessed the learning rate, expected future learning rate and the potential of the SSD in different visual tasks and navigation environments, as subjectively perceived by the participants. It also served as means to collect final feedback in the form of freely answerable questions. The first part of the evaluation uses a structure based on a System Usability Scale [47], allowing participants to rate the different aspects on a scale from 1 (very low) to 5 (very high).”

Publications added:

• J. Brooke – 2006 - SUS - A quick and dirty usability scale

Comment: Line 313 onwards (A): "within ten hours of training, the SSD could not outperform the white cane as navigational aid. This finding is expected, as it is consistent with previous studies." This statement puts in question why you would spend all this time and effort developing something when you do not expect that it will improve on the current clinical standard of care – the while cane. The authors need to frame this fact in order to explain better why the study was conducted in the first place.

Response: Thank you for drawing attention to the misleading phrasing of this passage. The fact that the SSD itself could not outperform the white cane was indeed to be expected, however, it was not the focus of the research question. The goal of the SSD is to generally improve the navigation performance of blind persons, which does not exclude the additional use of a white cane in combination with the SSD. 

We assumed that if participants were only asked to navigate using either the white cane or a combination of both SSD and white cane, they would be less encouraged to use the information given by the SSD and instead purely rely on the familiar tactile information given by the white cane. In order to familiarize them with the new auditory information and its application for navigation, it was decided to include trials in which the SSD is the only form of navigational aid. These trials were only measured and evaluated since they give good insight into the potential and limitations of the SSD, for example whether it is possible at all to detect obstacles and walkable paths. This text passage was changed in the revised manuscript, to better convey the implications of the SSD-only trials and the SSD + white cane trials.

Changes in the manuscript (lines 373-385):

• “This finding is expected for SSD-only trials, as their main purpose in the study was to train and familiarize participants with the new type of navigation aid without the option to ignore it and rely solely on the white cane. It is also consistent with previous studies of similar devices [37]. The reason is that all participants already had years of training with the white cane, whereas the SSD poses as a new and unfamiliar experience. However, the hypothesis of the SSD being able to improve the overall navigation performance of participants when combined with a white cane, could also not be shown within the duration of training. This is shown by the navigation performance with SSD+cane not significantly varying from those with just the cane, both in average collisions and required navigation time. While this may be, as will be later discussed, only a result of lack of training rather than a showcase of the absolute limit of the device, it certainly indicates that the device cannot be seen as instantaneous and learning-free improvement to standard navigation with a white cane.”

Comment: Line 313 onwards (B): In comparison, the object identification data demonstrate the potential use of the device, as that would go above and beyond what a cane can do. The authors need to maximize this finding. – Interesting that the participant data did not support this result, as the participants immediately realized that this practical implication is not all that practical. So there is some work to be done to improve this potential?

Response: We appreciate your comment. We expanded the discussion part about potential adjustments and optimizations of the SSD on the object recognition functionality (lines 479-485).

We hesitate to shift the focus too much on the object recognition experiments. This application field of the SSD was not our priority during the development of the SSD and the design of the study. Thus, it is less elaborated than the navigational aspects. This is, for example, shown by the fact that there were no randomized and repeated object recognition trials of the same objects that would allow to analyze the results for any learning effects. We rather see the findings of this study as a motivation for a new study that further elaborates on the application of the SSD for object recognition, perhaps with direct comparison to other SSDs. To convey this last point, we added another text passage to the discussion part (lines 411-416). (Note: EASV is the name of the SSD developed within this study. A more detailed explanation is found in the last point of the response to the second reviewer.)

Changes in the manuscript (lines 414-419):

• “These findings suggest a follow-up study with stronger focus on object recognition using the EASV, including training phases and feedback questions aimed more towards this function and its usefulness in everyday life. Such a study allows to assess potential reasons for the difference between objective results and subjective rating of object recognition and whether the occurrence of this difference is limited to the first trials or persists even through longer training periods.”

Changes in the manuscript (lines 482-488):

• “Further, to support especially the object recognition function of the device, the color or surface structure captured by the center field of the visual input image could be translated into different variations of non-monotone sounds. A similar method of color translation, using different instrumental sounds, was explored and successfully applied for object recognition tasks with the EyeMusic SSD [32]. However, such an implementation of additional types of auditory stimuli must be carefully weighed up against the increased information load on the auditory sense.”

Comment: Line 372 – "Thanks to the close collaboration with blind persons" would indicate that this was a participatory action research? Did the authors actually consult with blind persons BEFORE conducting the study, or were they involved in the design of its priorities, questions to be asked, etc.? Were they involved in the analysis and interpretation of the data or the preparation of this manuscript. These aspects would make it a close collaboration. Please clarify.

Response: Thanks for pointing this out. There was a pre-experiment consultation of one blind person to showcase the device and change aspects, such as the sound output of the device (from instrumental to non-monotone burbling sounds, since instrumental sounds were perceived as distracting) and a more distinct indicator for the perception of vertical position was implemented due to the suggestion of the blind participant. However, the phrase mentioned here indeed referred to the participants of the study and thus, "collaboration" is the wrong term. We changed the respective text passage from "Thanks to the close collaboration with blind persons, this study revealed […] "to "The participant feedback on the SSD that was gathered in the last two questions of the evaluation revealed […] ", found in line 461 in the revised manuscript.

Comment: Mention in limitations: The full-field virtual set-up eliminates the use of any remaining visual field. Is this a worth price paying for what is gained by the device? 

Response: Great question. Covering up any remaining visual field with the SSD is indeed a major problem of the SSD that would have to be addressed before the SSD could be considered for real-world usage. It is a result of technical hardware limitations which to circumvent would have gone beyond the scope of the study and current expertise of the authors. The problem and a possible solution are indirectly addressed in line 378 in the old manuscript version as part of the obtrusiveness of the hardware. We understand that this point should be mentioned more specifically and changed the phrase from "First, the overall hardware of the device must be changed in order to be more comfortable, subtle and less obtrusive, especially when considering the development of a prototype for use in everyday life." to "First, the overall hardware of the device must be changed in order to be more comfortable, subtle and less obtrusive and also in order to no longer cover up the visual field, allowing to fully use any remaining vision. This is especially important when considering the development of a prototype for use in everyday life.". The change is found in line 465 of the revised manuscript.

For the experiment itself, we do not think that the elimination of the use of the remaining visual field impacts the overall findings of the study. Contrary to a real-world application, the goal of the study is not to improve the navigation performance of participants to its absolute maximum, but instead to compare the navigation performance with SSD to the performance without SSD under otherwise same conditions. These "otherwise same conditions" include the fact that in both scenarios, the remaining visual field of the participants was covered, as was further discussed in the response to line 140(B).

In response to the second review:

Comment: There are only 7 participants in the study. Even for a study comparing completely blind, congenitally blind participants, this is a very small sample. According to the description of your participants given in the methods section however, most of your participants would be considered low vision (or late blind) according to the definition that is usually given in the literature. Congenitally blind (CB) participants with no light perception are indeed very hard to find, and sometimes a sample of only 10 CB can be acceptable. In the case of low vision participants, it is somewhat relatively easier to find participants, thus I really think this sample of only 7 participants is inadequate. Can you possibly increase the sample size for this group to 10 participants minimum?

Response: We completely agree that the sample size of the study is very small and may even be one of the main reasons why the significance threshold of some hypothesized effects was not met. Unfortunately, given the relatively high time expenditure and especially with the current corona situation, it is unlikely to find more participants in a reasonable amount of time. Nevertheless, we believe that the study gives valuable insights in the therapy of blind people with SSDs. 

The study took place in Marburg, which is the city with the highest percentage of blind persons in Germany. The call for participants was send via the university's service center for blind students as well as the magazine of the DVBS ("German Association for Blind Persons in Studies and Career"). Although not very high in comparison to study populations, the number of seven blind persons interested in participating in the study was by far the highest out of the three regions, Marburg, Tübingen and Düsseldorf, in which the call for participants was send out. Only one person in the area of Tübingen expressed their interest in participating in the study and two persons in the area around Düsseldorf. The person located in Tübingen was consulted for feedback during the development of the SSD, but due to time limitations, it was decided to only carry out the study with the seven participants in Marburg. Furthermore, the availability of the seminar rooms in which the study took place was quite limited, as they were used for normal lecture schedules as well. All of these factors unfortunately led to the low number of participants.

Comment: The lack of control group in the study. As I mention in my previous comment, there are only 7 participants in the whole study... and no control group. A control group of blindfolded sighted individuals could have been added to this study making teh results much more interesting. I understand that participants served as their own controls in the study, the white cane condition vs. the SSD condition vs. the combined condition, but this comparison only makes sense if you compare it with a group of participants with no experience with the white cane, as you correctly mention in your discussion. You could have disentangled the issue of experience with the white cane. Furthermore, it is generally accepted that CB participants who are completely blind with no light perception benefit from mechanisms of brain plasticity that enable them to recruit visual brain areas for tasks involving other modalities (for example via SSD). This usually affords them better behavioral performance with these devices. Thus, it is unclear if CB participants could have performed better than this group of low vision participants in the current study.

Response: Highly appreciated comment. A sighted control group – although not representative for the potential navigation performance with an SSD in general – may have given similar insight in the comparison between white cane and SSD. If any future studies with the SSD are realized, the addition of a control group with no experience with either navigational aid will certainly be one of its main aspects. However, as mentioned in the point above, the number of possible participants for this study was limited due to time constraints and limited availability of the study location. Thus, it was decided to focus on actual blind participants, as it can be sometimes criticized when studies regarding blind persons are carried out with blindfolded sighted participants instead.

We added a text passage to now mention the possibility for a control group of blindfolded sighted participants. Further, two new citations regarding the difference of blind and blindfolded participant performance in navigation and recognition tasks were added.

Changes in the manuscript (lines 388-393):

• "However, in a practical setup such a limiting participation criterion is not feasible, as about 80% of blind or legally blind persons have experience with a white cane. With blindfolded sighted participants, this lack of participants not familiar with the white cane could be tackled. However, multiple studies found that results of blindfolded participant performance in navigation and recognition tasks are not representative of the performance of blind participants."

Publications added:

• A. Kolarik – 2017 - Blindness enhances auditory obstacle circumvention: Assessing echolocation, sensory substitution, and visual-based navigation

• L. Li – 2002 - Heading perception in patients with advanced retinitis pigmentosa

Comment: Participants get better at a task each time they do it. Your study compared participants with SSD vs. the same participants with SSD and cane, the same participants again with only a white cane. It is unclear what measures were taken to circumvent the repetition effect. Did participants do the tasks in different orders, enabling you to compare performances between those subjects who did the cane first part and then the SSD part and the combined part? This is unlikely given the small sample size! If no measures were taken, or worst, if all participants did the SSD part first and the cane part of the study, your effects might simply be due to this repetition effect. Could you please explain what was done to control for this? 

Response: Thanks for asking. The condition (SSD, white cane, SSD + white cane) was randomized for each trial individually. There were no "parts" in which only one condition was tested. We see that the phrase in lines 185-187 of the old manuscript version "The tasks were done in randomized order to minimize the influence of potential learning effects when comparing trials of different conditions." did not properly convey the fact that randomization occurred in-between each trial and not just between parts consisting of multiple trials. The sentence was changed to "All trials were done in randomized order independent of their condition to minimize the influence of potential learning effects when comparing the trials of all three conditions.", found in line 236 of the revised manuscript.

Comment: Lack of knowledge of the literature. Although you do cite some key studies in your Introduction, you do not cite THE key studies that would be relevant to this study. Experiments on object recognition and spatial navigation, reviews on brain plasticity and SSD use, reviews offering explanations why SSDs are not commonplace. You also sometimes cite teh wrong studies, for example you cite teh concept of brain plasticity and offer a citation, but this study is not the one that coined the term brain plasticity. Also, there have been several studies using SSDs for obstacle avoidance and navigation, and these studies are not discussed. Other studies have shown that it is possible to use a SSD to navigate as efficiently as sighted participants using vision in simple environments, or that in certain situations, SSDs can give you information that enables to navigate more efficiently than a white cane. There are even studies that have used virtual reality to train participants, and that transfer between real and virtual environments was possible for participants of varying degrees of visual expé rience, which could be an interesting point for the discussion of yoru results. The fact that you omit these important papers gives the impression that you are unaware of these studies.

Response: Thanks for the advice. We cited the publications we assessed as best conveying the key points relevant to the respective topics. However, we thank you for bringing to our minds the advantages of citing more publications, especially for the key statements of the study, to cover the different fields of interest the readers come from. We can also not exclude the possibility to have missed important literature during our research. If you think that essential publications are missing even after the inclusion of the following list of added citations, we would be thankful for a reference link or the names of the specific studies. 

The following list of added citations and resulting changes of text passages is sorted by topic:

Experiments on object recognition and spatial navigation

In our manuscript introduction and also in the comparison of our results to the literature, we chose to focus on three SSDs and respective studies: two of the most popular SSDs, the vOICe and EyeMusic, as they are to this day topic of many studies and are referenced in nearly all publications regarding the topic of SSDs, and the Sound of Vision (SoV) system that is, to the best of our knowledge, the most extensive and advanced SSD, based on the technological features it provides and the number of research groups involved in its development.

To increase the range of SSDs covered in this paper, we added the visual-to-tactile haptic belt "ALVU" and re-introduced the single direction, distance-based "EyeCane", which was already included in the original thesis paper, to the summary in the introduction. Additionally, the study using the EyeCane is cited in the literature comparison. These devices were originally excluded from the manuscript, as their design suggests the application as a substitute for the white cane rather than as a complementary device. Seeing how they can still offer very similar functionality to the SSD developed in our study in terms of navigation and that they are based on distance translation, we reconsidered that decision in view of your feedback.

Changes in the manuscript (lines 91-115):

• “Some newer devices, such as the SSDs or ‘EyeCane’ [38], ‘Sound of Vision’ (SoV) [39], ‘Array of Lidars and Vibrotactile Units’ (ALVU) [40] use depth as their primary visual input. Using this information, they directly translate the visual distance into either tactile or auditory feedback. 

The EyeCane is a hand-held device that captures depth in a single-direction and translates the distance into vibration strength. This results in the device functioning very similar to a standard white cane, however, it offers the advantage of increased detection range of up to 5m [41]. 

The SoV uses a combination of stereo camera and infrared sensor, both worn on the forehead, to capture a depth information array of the scene and translate each point of the array into the volume of an auditory output signal [42]. The auditory output of the array happens simultaneously, resulting in a high update rate, which allows for an easier understanding of the 3D shape of the environment. The SSD however requires a multitude of peripheral devices – multiple cameras and depth sensors and a laptop carried in a backpack – which makes it not only impractical and inconvenient in mobile scenarios such as navigation, but also increases the cost of the device. 

The ALVU consists of an array of seven infrared depth sensors fixed to a wearable belt that capture the distance to obstacles and surfaces in an ±70° horizontal and ±45° vertical angle in front of the user, translating the information into vibration strength of a vibratory motor array. This allows the perception of obstacles and surfaces in a horizontal arc, similar to the information gathered by sweeping motion with a white cane, and additionally allows to perceive obstacles at waist or head level. This SSD was found to have a low learning barrier and allowed navigation performance similar to a white cane. However, due to the amount of individual sensors and vibratory motors, the cost sums up to around 1300$ [40]. Given that 89% of visually impaired people live in low- or mid-income countries [43], a low-cost solution for SSDs is crucial.”

Changes in the manuscript (lines 441 to 451):

• “In a navigation study with both real and virtual maze environments, the navigation performance of the participants using the EyeCane was found to significantly improve, with both the average required time to navigate through the maze and the average collisions per trial being reduced to around half after 3 sessions of 90 to 120 minute training [52]. However, in the group of low vision and late blind participants, which is most comparable to the study population of the EASV study, the time required to solve the maze was 152s on average. That is significantly higher than the average required navigation time measured for the SSD-only trials with the EASV of 68±55s, despite an overall shorter distance from start to finish. While these differences in time may originate simply from differences of the experimental setup, such as the complexity of the obstacle course, it further suggests a low entry barrier of the EASV.”

Publications added:

• D. Chebat – 2015 - Navigation using sensory substitution in real and virtual mazes

• S. Maidenbaum – 2014 - The 'EyeCane', a new electronic travel aid for the blind: Technology, behavior & swift learning

• R. Katzschmann – 2018 - Safe local navigation for visually impaired users with a time-of-flight and haptic feedback device

We further added three publications in line 83 of the revised manuscript that describe the current state of Sensory Substitution Devices and summarize many projects in this field to cover those SSDs not mentioned specifically in our manuscript.

Publications added:

• M. Gori – 2016 - Devices for visually impaired people: High technological devices with low user acceptance and no adaptability for children

• A. Chaudhary – 2019 - State of Art on Wearable Device to Assist Visually Impaired Person Navigation in Outdoor Environment

• W. Elmannai – 2017 - Sensor-Based Assistive Devices for Visually-Impaired People: Current Status, Challenges, and Future Directions

Brain Plasticity

The publications originally cited in our manuscript in regard to brain plasticity were selected as they are specified on brain plasticity in sensory substitution. The publication' Bach-y-Rita – 1987 - Brain Plasticity as a Basis of Sensory Substitution' is one of the first publications in which brain plasticity is explained in its correlation to sensory substitution. The second publication on this topic, 'Bach-y-Rita – 2003 – Sensory substitution and the human-machine interface' was included to give a more modern view and description on brain plasticity in Sensory Substitution.

With regard to your feedback, we added the publication that, to the best of our knowledge, first coined the term "brain plasticity" (Bennett, 1964), as well as another, more recent publication (Cecchetti, 2016) to include findings from the more recent years. Further, we included the term "compensatory plasticity", as it has a similar meaning to "brain plasticity in sensory substitution", and cited a publication connecting that term to sensory substitution (Rauschecker, 1995).

Changes in the manuscript (lines 67-70):

• “It is possible that the substituting sense leads to very similar activities in the brain as the original sense would, even in cases of congenital sensory disabilities - a process called ‘brain plasticity’ [7,10–12] or, in the context of sensory substitution, ‘compensatory plasticity’ [13].”

Publications added:

• E. Bennett - 1964 - Chemical and anatomical plasticity of brain

• L. Cecchetti - 2016 - Are supramodality and cross-modal plasticity the yin and yang of brain development? From blindness to rehabilitation

• J. Rauschecker - 1995 - Compensatory plasticity and sensory substitution in the cerebral cortex

Challenges of sensory substitution

In the discussion of current limitations and challenges of the SSD and potential reasons for their lack of popularity, found in line 464 of the revised manuscript, we included two additional citations:

• S. Real – 2019 - Navigation systems for the blind and visually impaired: Past work, challenges, and open problems

• Á. Kristjánsson – 2016 - Designing sensory-substitution devices: Principles, pitfalls and potential 

Transfer between real and virtual environments

One of the studies already cited (Hoffmann, 2018) as well as the re-introduced publication about the EyeCane (Chebat, 2015) include training in a virtual environment. However, we are not certain if there really is a need to put any focus on the difference between training and experiments in virtual and real environments. Our setup simultaneously combines the features of both virtual and real environments with its real-time synchronization of digital and real-world obstacles. Since there are no individual navigation trials in either only a real or virtual environment, we cannot draw any conclusions about the differences between the two environment conditions and how they would affect the results.

Comment: Conclusions. Your results simply do not support your conclusions! You say some interesting this ion the discussion section about the future of SSDs , and where to place the camera, etc, but nothing in yoru results suggest these conclusions, even remotely. The Introduction and Discussion would be better by focusing on the themes of this study: Navigation and object recognition with a novel SSD. 

Response: Thank you for pointing out this flaw in the structure of our manuscript. In response to this feedback, we created a new, separate conclusion that focuses purely on statements directly supported by the results of our study. We still see value in mentioning potential improvements to the design of the SSD and included it as part of the discussion. 

Changes in the manuscript (lines 506-525):

• “Conclusion

In this study, the feasibility and performance of a spatial-information-based SSD was evaluated in a navigational task, using a simulated environment synchronized with real-world objects. The performance of the SSD was benchmarked against the performance with a white cane and with a combination of both SSD and white cane. In addition, the ability to correctly identify 3D objects and structures using the same device was assessed by presenting virtual objects and a selection of four answers. The participants’ feedback and perception of their learning progress was assessed in a concluding evaluation.

It was found that all participants were able to successfully navigate using the SSD, even immediately after the instructions. Without any prior training in object recognition and only three sessions using the SSD, participants recognized complex 3D objects such as cars, chairs and staircases when presented in otherwise empty environments. Further, this was done using a consumer-level smartphone, showing the technological feasibility of depth-based sensory substitution without the use of expensive peripheral devices.

However, the study also reveals the limitations of the developed SSD. Within the duration of the training, the SSD could offer no significant advantage compared to navigation with the white cane, even when both these navigational aids were used in combination. Further, while object recognition did show a significant success rate, the average time of almost 100 seconds that was required to recognize one object makes this feature impractical for real-world application without more extensive training.”

Comment: Learning rate. The main thing that you claim to be testing in this study is learning rate. It is unclear how your compared performances to asses if indeed learning had taken place. Usually comparisons between early training and late training is assessed, or comparisons between novies and experts. Is the only way that you assessed training by self-report? 

Response: The potential training effect was analyzed using a linear model for the correlation of training (where training is measured by the number of trials already absolved) and the performance (individually assessed through both the time required to finish the navigation trial and the number of collisions per trial). Due to the already small sample size, all navigation trials simultaneously act as both training and as measurement trials.

We did explore the possibility of doing a two-way-Anova analysis, comparing performance only in early vs late trials (first 5 or 10 trials vs last 5 or 10 trials). However, this would have reduced the sample size to one or two thirds of the total amount and did not lead to significant results. This made the linear model the better option to analyze the results, as it includes all navigation trials and thus maximizes the sample size. Furthermore, even if training and measurement phase would have been separated in the study design, it would have led to the same problem of decreased sample size, given that the total duration of the study could not be increased. We specified the text passage regarding the statistical evaluation of the navigation performance in line 284 of the revised manuscript to now state "[…] allowing to test for significant learning effects over the duration of the study in any of the conditions."

Statements made towards the learning rate of participants using the SSD are solely based on the results of the navigation, indicated in form of objective parameters (required time per trial and number of collisions), unless the self-report of the participants as part of the final evaluation is specifically mentioned.

Comment: Figures. Need to be prepared with Illustrator or photoshop. The copied images into the text look grainy and not very clear (ex. Fig 1, Fig 6, 7, 11, 12, &13). There is no reason for images with text to be grainy. There are alot of figures. Maybe you could put methods figures in one ,a nd the results figures together? Might I suggest making one single figure or possibly even only 2 for the methods ? Making one figure from figures 1-6 could really help clarify the information and make it more succinct. The results figures need some work also and are very grainy (low resolution) and not very clear. Maybe a bar graph would be more adequate?

Response: Great advice. The figures 9 and 10 were re-made in higher resolution. Following the recommendation of the first reviewer, figures 11, 12 and 13 were removed and the results of the evaluation were instead included as percentages in the text passage.

To decrease the number of figures further, we combined figures 1, 2, 3, figures 4, 5, 6, as well as figures 7 and 8. The resolution of figure 6 (now figure 2(C) ) was increased.

Comment: Methods. Not enough information is given about teh device itself, its field of view and method of translating visual information. Maybe a video file uploaded as supplementary material would be helpful? Also, it might be useful, at least for the sake of this paper, to give your SSD a name ! The Central Sound View? Because it translates the central visual field into sounds?

Response: As recommended, we added additional information on the functionality of the SSD. Regarding the video file, there should be a video file uploaded as 'S6 Video. Navigation showcase' which shows a side-by-side comparison of the participants "first person view" (the video captured by the visual input camera of the SSD, including the virtual obstacles) and a third person view of the same navigation trial.

Thank you for suggesting a name for our SSD. However, we do not see the centralized visual information translation as the main feature of the device but rather as a result of the attempt to lower the introduction barrier of the SSD. The most distinct aspects of our SSD, in our opinion, are the implementation of a distance-based translation algorithm without any complex peripheral devices as well as its low introduction barrier. Thus, we chose to name our device EASV, 'Easy-Access SoundView' and replaced the phrases "SSD developed within this study" with the name wherever possible.

Changes in the manuscript (lines 156-166):

• “The vertical angle of the visual input field was set to 45° to align with the vertical viewing angle of the smartphone camera. In the horizontal axis, the angle of the visual input equals 0° due to the single-column design of the translation algorithm (Fig 1(B) ). The device tracks the distance to obstacles and surfaces in 9 different directions along the vertical axis, which results in a resolution of 5.6°. 

The information about the distance to the next surface measured by each individual ray is translated by the SSD into non-monotone sounds similar to burbling water. Each ray has its own respective pitch depending on its vertical direction, starting at around 143Hz for the lowest ray up to 880Hz for the highest.

As this SSD is aimed towards ease of access in terms of both training and hardware requirements, it was named Easy-Access SoundView (EASV).”

Comment: It is not clear from your methods section how object recognition was done. Were they simply recognizing the obstacles in the obstacle course?

Response: The object recognition task was carried out using separate scenes. While the only objects found in the obstacle course are the large rectangular shapes of the boxes, the objects in the object recognition trials were more complex and were presented without any surrounding environments. To make this more clear in the text, the sentence in line 252 of the revised manuscript was changed from "In the object recognition trials, participants were presented different virtual objects […] "to "In the object recognition trials, participants were presented different virtual objects in individual, otherwise empty scenes […]". 

Other changes:

• We changed the headline to “Navigation Aid for Blind Persons by Visual-to-Auditory Sensory Substitution: A pilot study”. We think pilot study fits really well to the publication. 

• We changed the acknowledgments to no longer state "Funding received from University of Tuebingen (ZUK 63) as part of the German Excellence initiative from the Federal Ministry of Education and Research – Germany (BMBF). This work was done in an industry-on-campus-cooperation between the University of Tuebingen and Carl Zeiss Vision International GmbH.". This text passage was instead moved to the funding section of the submission form. Instead, a thanks to participants and people who helped during the development of the device was added.

We again thank both reviewers for the detailed and valuable feedback. We hope that this response could address all issues sufficiently and that the changes made to the manuscript coincide with the improvements to our work intended by your suggestions.

Best regards,

Alexander Neugebauer

---

## [Editor Report · Decision Letter 1]

24 Jul 2020

Navigation aid for blind persons by visual-to-auditory sensory substitution: A pilot study

PONE-D-20-09308R1

Dear Dr. Neugebauer,

We’re pleased to inform you that your manuscript has been judged scientifically suitable for publication and will be formally accepted for publication once it meets all outstanding technical requirements.

Kind regards,

Maurice Ptito

Academic Editor

PLOS ONE
---

## [Editor Report · Acceptance letter]

29 Jul 2020

PONE-D-20-09308R1 

Navigation aid for blind persons by visual-to-auditory sensory substitution: A pilot study 

Dear Dr. Neugebauer:

I'm pleased to inform you that your manuscript has been deemed suitable for publication in PLOS ONE. Congratulations! Your manuscript is now with our production department. 

Kind regards, 

on behalf of

Dr. Maurice Ptito 

Academic Editor

PLOS ONE